# Synthesis, Docking, and Machine Learning Studies of Some Novel Quinolinesulfonamides–Triazole Hybrids with Anticancer Activity

**DOI:** 10.3390/molecules29133158

**Published:** 2024-07-02

**Authors:** Krzysztof Marciniec, Justyna Nowakowska, Elwira Chrobak, Ewa Bębenek, Małgorzata Latocha

**Affiliations:** 1Department of Organic Chemistry, Medical University of Silesia, Jagiellońska 4, 41-200 Sosnowiec, Poland; s85039@365.sum.edu.pl (J.N.); echrobak@sum.edu.pl (E.C.); ebebenek@sum.edu.pl (E.B.); 2Department of Molecular Biology, Jagiellońska 4, 41-200 Sosnowiec, Poland; mlatocha@sum.edu.pl

**Keywords:** machine learning, molecular docking, ADMET, quinolinesulfonamides, triazoles, anticancer activity

## Abstract

In the presented work, a series of 22 hybrids of 8-quinolinesulfonamide and 1,4-disubstituted triazole with antiproliferative activity were designed and synthesised. The title compounds were designed using molecular modelling techniques. For this purpose, machine-learning, molecular docking, and molecular dynamics methods were used. Calculations of the pharmacokinetic parameters (connected with absorption, distribution, metabolism, excretion, and toxicity) of the hybrids were also performed. The new compounds were synthesised via a copper-catalysed azide–alkyne cycloaddition reaction (CuAAC). 8-*N*-Methyl-*N*-{[1-(7-chloroquinolin-4-yl)-1*H*-1,2,3-triazol-4-yl]methyl}quinolinesulfonamide was identified in in silico studies as a potential strong inhibitor of Rho-associated protein kinase and as a compound that has an appropriate pharmacokinetic profile. The results obtained from in vitro experiments confirm the cytotoxicity of derivative **9b** in four selected cancer cell lines and the lack of cytotoxicity of this derivative towards normal cells. The results obtained from silico and in vitro experiments indicate that the introduction of another quinolinyl fragment into the inhibitor molecule may have a significant impact on increasing the level of cytotoxicity toward cancer cells and indicate a further direction for future research in order to find new substances suitable for clinical applications in cancer treatment.

## 1. Introduction

The interest in quinoline derivatives as medicinal substances comes from research on natural compounds, i.e., alkaloids present in the bark of the cinchona tree (*Cinchona* L.), such as quinine, which has antimalarial activity, and its dextrorotatory stereoisomer, quinidine, which has antiarrhythmic activity. In an effort to obtain more effective and less toxic antimalarial drugs, many synthetic derivatives of 4- or 8-aminosubstituted quinoline have been obtained, including chloroquine, amodiaquine, primaquine, and mefloquine. However, 8-hydroxyquinoline derivatives, such as chlorquinaldol, clioquinol, broxychinoline, and broxaldine, have antibacterial and antifungal properties [1]. 4-Quinolone derivatives, e.g., moxifloxacin, also have a strong antibacterial effect.

Numerous natural and synthetic quinoline derivatives also exhibit anticancer activity. This group of substances includes, among others, topoisomerase I inhibitors—camptothecin and its semisynthetic derivatives topotecan and irinotecan [2,3]; and tyrosine kinase inhibitors—bosutinib, lenvatinib, cabozantinib [4,5,6], and tipifarnib [7]. Many new compounds containing the quinoline system have been described as having high cytotoxicity, inhibiting lactate dehydrogenase, pyruvate kinase, mitotic kinesin-5, thymidylate synthase, carbonic anhydrase, telomerase, aromatase, sirtuin, and protein kinase enzymes and inhibiting tubulin polymerisation, free radical regulation, apoptosis, iron chelation, etc. [8,9,10,11,12].

The analysis of data published in the medical and pharmaceutical literature indicates that in recent years, there has been an increase in interest in the derivatives of quinoline sulfamoyl (sulfonamide) [13,14,15,16,17]. Despite their often-simple structure, these compounds are characterised by a wide spectrum of biological activity: anticancer, antidepressant, antiviral, and analgesic. The anticancer activity of the sulfonamide quinoline derivatives is based on the inhibition of lactate dehydrogenase, pyruvate kinase, tubulin polymerisation, carbonic anhydrase, and RhoA/ROCK (Rho-related coiled–coil kinase). It should be mentioned that a large group of ROCK inhibitors contains an azine system in its structure, including, among others, pyridine, isoquinoline, quinoline, or quinazoline. And some of the sulfamoyl derivatives of azines, which are ROCK inhibitors, have been approved in Japan and/or China (fasudil and ripasudil) [18,19,20,21,22,23,24,25] (Figure 1).

Rho-associated protein kinase is one of the best characterized effectors of the small GTPase RhoA and belongs to the serine/threonine AGC family of protein kinases, which also includes kinases A, G, and C (PKA, PKG, and PKC) [27]. The ROCK family consists of two isoforms, ROCK1 and ROCK2, sharing 65% overall homology and 92% homology in the kinase domain. Both kinases contain a catalytic domain at the N-terminal domain followed by a central coiled–coil domain that includes the Rho-binding domain (RBD) and a C-terminal pleckstrin–homology (PH) domain. The ROCK family plays a central role in diverse cellular events, including gene expression, the regulation of cell detachment, cell movement, and the establishment of metastasis. The overexpression or dysfunction of ROCK would lead to diseases such as hypertension, stroke, diabetes, glaucoma, and neurodegenerative diseases [28,29,30,31]. Currently, the only ROCK inhibitor clinically approved is fasudil (Figure 1), which has been used safely in Japan since 1995 for the treatment of a cerebral vasospasm after a subarachnoid hemorrhage (SAH). Although, fasudil is a more potent Type I inhibitor of ROCK (IC_50_ = 1.2 and 0.82 μM for ROCK1 and ROCK2, respectively) relative to related AGC family kinases, it is a non-selective ROCK kinase inhibitor drug [32,33]. There are many scientific reports confirming the thesis that ROCK plays an important role in tumour development, progression, and metastasis. Many ROCK inhibitors have been investigated as potential inhibitor therapeutic substances in the treatment of cancer, including bladder cancer and hepatocellular carcinoma melanoma, eye cancer, glioblastoma multiforme, prostate cancer, and pancreatic ductal adenocarcinoma (PDAC) [34,35,36,37,38,39,40].

Based on published crystallographic data and a computer-based model of the ligand–enzyme interaction model, most reported ROCK inhibitors consist of 3 moieties (Figure 2) [41]. Part A is the nitrogen-containing aromatic ring, such as azines (isoquinoline, quinoline, and pyridine); pyrazole; or azaindole. Part B, which is a link connector of parts A and C, or is integrated directly or by means of an amide or sulfonamide group, is structurally differentiated. It may contain a thiophene, thiazole, triazole, and tetrazole aromatic system. Part C contains a very diverse number and type of moieties, including the triterpene system [42,43].

Virtual screening (VS) is one of the popular techniques for searching for potentially bioactive molecules from input chemical libraries. Structure-based virtual screening (SVBS) is of great interest in the study of new mechanisms such as ligand–protein interactions, especially in the era of searching for new bioactive substances based on artificial intelligence (AI) [44,45,46]. In SBVS studies, compounds are placed first in the binding pocket and then scored. A key problem is scoring, which results in a large number of inactive compounds ranking high on the scoring list or a loss of active molecules. There are many sources of error in scoring, including difficulties in implementing complex energy terms in fast scoring functions and, of course, related challenges in accounting for protein flexibility, among others [47,48,49]. Several strategies have been proposed to improve this process. One of them is machine learning (ML) [50,51,52]. In this work, we used machine learning to solve these problems. ML compensates for the shortcomings in rigorous theory by learning from the data of known experiments. This approach can improve the ability of the ensemble docking to classify compounds as active and inactive, and performance does not decrease as more structures are added to the ensemble [53].

The basic idea of our machine-learning approach is to use the multiple docking scores of each compound to the ensemble of structures as features in machine-learning models [54]. By learning how active compounds differ from inactive compounds in these features, the machine-learning approach improves predictive performance. The machine-learning approach has also removed the problem of potentially worsening rather than improving performance by having just beyond a few structures [53].

The purpose of this work was to design, using machine learning, and synthesise hybrids of 8-quinolinesulfonamide and 1,2,3-triazole (Figure 2). Compounds containing the 1,2,3-triazole system exhibit a wide spectrum of biological activity, including anticancer activity. Therefore, in this work, we decided to combine the systems of quinoline sulfonamide and triazole to obtain derivatives with anticancer activity [55,56,57].

## 2. Results and Discussion

### 2.1. Design and In Silico Prediction of ROCK1 Inhibitors

#### 2.1.1. Machine Learning

The aim of this work was to discover new hybrids of quinolinesulfonamide and triazole with anticancer activity and the potential mechanism of action as Roh inhibitors. From an in-house library of quinolinesulfamoyl compounds (see Appendix A), 22 derivatives were selected for synthesis (Figure 3) using ML approaches.

In this part of the in silico research, the EDock-ML web server was used. The basic idea of the EDock-ML machine-learning approach is to use the multiple scores of each compound to the ensemble of structures as features in machine-learning models. A K Nearest Neighbour predictive model was used in this research [53]. Results obtained from the EDock-ML sensitivity parameter indicate whether the selected machine-learning model predicts that the relationship will be active with a probability of 90% or whether the model predicts that the relationship may be inactive with a probability of 90%. As can be seen from the data presented in Table 1, all tested compounds have a probability of activity of 90%. The best specificity of 0.9 in the selected model is demonstrated by six compounds (compounds **5a**, **8a**, **9a**, **9b**, **11a**, and **14b**). However, the lowest specificity parameter of 0.56 was obtained by four derivatives (compounds **7a**, **7b**, **11b**, and **12b**), which means that the K Nearest Neighbour model could certainly predict the compound to be active with 56%. This probability is not as high as 90% but is still good.

The last parameter generated by the predictive model is the AUC. AUC stands for the Area Under Receiver Operating Characteristic Curve. Each value varies between 0 and 1, with 1 giving the best possible model. A value of 0.5 means that a model performs only as well as a random model. The closer the AUC value is to 1, the more reliable the machine-learning model is. From Table 1, we see that the model predicts the compounds **4**–**14** to be active with 82% probability, which means that the probability of these compounds being active was high. To summarize this part of the research, it can be stated that the highest values of sensitivity, specificity, and the AUC parameters were obtained by compounds **5a**, **8a**, **9a**, **9b**, **11a**, and **14b**, which indicates a high probability of their activity towards the selected target. It should be noted that the remaining derivatives also obtained good probability results.

#### 2.1.2. Molecular Docking

In the next stage of the in silico research, in order to determine the detailed type of interactions between the selected compounds and ROCK1, the compounds selected by ML were docked using the AutoDock Vina programme. We used the ROCK1 complex with fasudil (PDB ID: 2ESM). The derivatives of quinolinesulfonamide **4**–**14**, as ranked by Vina, are presented in Table 1. The lowest scores correspond to a strong binding affinity and the most likely ligand–protein system. Comparing the docking-score values obtained for 2ESM, it can be concluded that the vast majority of compounds **4**–**14** show a significantly higher affinity to fasudil. Compound **9b** shows the lowest ∆G value and the highest potential binding affinity to the target protein (−10.4 kcal/mol). It should be mentioned that derivative **9b** obtained high probability coefficients for interactions with the selected protein target in the machine-learning analysis. Compounds **10a** and **10b** also show high potential activity in in silico studies (Table 1).

Detailed data on the interactions of derivatives **9b**, **10a**, and **10b** with the complex target protein in the obtained results are presented in Figure 4 and Figure 5 and Table 2.

The 7-chloroquinolinyl fragment interacts through π–sigma interactions with Val90 and Leu205. Additionally, these interactions are stabilised by alkyl–alkyl and π–alkyl interactions with Ala103, Ala153, and Ala215 and van der Waals interactions with Val137, Met156, and Tyr155. It should be emphasised that this pattern of interactions is visible between the isoquinoline fragment in the fasudil molecule and the amino acids of the ROCK1 binding site (2ESM). In addition, interactions between aromatic triazole systems and the benzene ring of the sulfamoylquinoline with Lys105 and Asp216 are also visible substituents (π–cation and π–anion interactions). Furthermore, the aromatic system of the pyridine ring interacts with Phe120 π–π in the T-shaped orientation.

#### 2.1.3. Molecular Dynamics Calculations

In order to verify the results obtained using the molecular-docking method, calculations were performed using the molecular dynamics technique. The lowest energy complex obtained from the Vina programme was used as the input structure in these calculations, that is, the complex of derivative **9b** with the target protein (∆G = −10.4 kcal/mol). To verify the stability of the tested system, both proteins and ligands, RMSD values were calculated on the time scale. The RMSD values of the **9b**–protein complex are shown in Figure 6. The RMSD in the **9b**–2ESM complex increases to approximately 3 Å in the first ten nanoseconds. Then, after about 20 ns, the RMSD value drops to about 1.5 Å, and after another 10 ns of calculations, it again reaches a value of about 3 Å. Then, the RMSD value decreases to approximately 2 Å and remains at this level until the final simulation time.

As part of the analysis of the results obtained from the molecular dynamics calculations, the RMSF value was also determined (Figure 7). RMSF is a calculation of the individual residue flexibility, that is, how much a particular residue moves (fluctuates) during a simulation. RMSF can structurally indicate which amino acids in a protein contribute the most to molecular motions. The data obtained from molecular docking indicate (Figure 5, Table 2) that the amino acids that interact in the protein binding site are Val90, Ala103, Lys105, Phe120, Val137, Ala153, Tyr155, Met156, Leu205, Ala215, and Asp216. The RMSF values presented in the mentioned figure indicate that the acids are subject to slight fluctuations during the simulation, which contributes to the high stability of the complex tested.

Figure 8 shows the results of the analysis of intermolecular hydrogen bonds formed between the ligand and the protein during the simulation. The maximum number of hydrogen bonds that occurred simultaneously during 50 ns was 2. Qualitatively, four different intermolecular hydrogen bonds were detected between ligand **9b** and amino acids of the protein binding site: Val90, Lys105, Met156, and Leu205. Hydrogen bonding with Lys105 occurred for 3.02% of the calculation time and with Val90 for 1.74% of the time. The key 2ESM amino acid residues that form hydrogen bonds in the **9b**–2ESM complex are Met156 and Leu205 (for 6.30% and 10.82% of the calculation time, respectively).

The in silico analyses carried out indicate a high potential for **4**–**14** derivatives to modulate the activity of ROCK1.

#### 2.1.4. Prediction of Drug-likeness and ADMET Profile of Designed Compounds

The chemical structure of compounds affects such structural properties as molecular weight (MW), the number of rotational bonds (nROTs), the possibility of forming hydrogen bonds (nHBA and nHBD), lipophilicity (logP), molecular refractivity (MR), and the topological polar surface of the molecule (TPSA). The way in which a substance with a specific structure interacts with the physical environment results from its physicochemical properties, such as solubility or permeability. The degree of interaction of a compound with proteins depends on its biochemical properties (e.g., metabolism, transporter affinity, binding, and target affinity). The pharmacokinetics and toxicity of a chemical compound that is a potential drug candidate are the result of its interactions in the physicochemical and biochemical environment of living systems (clearance, half-life, bioavailability, and LD50) [51].

The starting point for analysing the drug similarity of the substances designed in this work was to check the extent to which the tested structures meet the rules formulated by Lipinski, Ghose, and Veber. For this purpose, parameters calculated using the SwissADME website were used (Table 3) [58].

Lipinski’s rule of five (RO5) defines the drug-likeness of a chemical compound with favourable physicochemical properties, whereby the compound has biological activity and is designed for an oral route of administration. This rule describes a drug candidate through appropriate values of parameters, such as lipophilicity (determined as MLOGP ≤ 4.15), the molecular weight of the substance (MW ≤ 500), and the number of hydrogen bond acceptors (Nhba ≤ 10) and donors (nHBD ≤ 5) [53]. Lipinski’s rules recommend that an orally bioactive drug should not have more than one violation. As shown in Table 3, the molecular weight of compounds **4a**–**14a** and **4b**–**14b** range from 389.43 to 828.11 g/mol. Compounds **10b**, **11a**, **11b**, **14a**, and **14b** do not meet the molecular weight criterion. The MLOGP values for derivatives **4**–**13**, containing both a primary sulfonamide group (series a) and a secondary sulfonamide group (series b), regardless of the type of substituent in the triazole ring, are within the range predicted by Lipinski’s rule. Only both triazole derivatives substituted with a large triterpene hydrophobic system, **14a** and **14b**, are characterized by high lipophilicity (MLOGP equal to 5.14 and 5.30, respectively).

Ghose defined an organic compound that is a drug-like molecule as one for which the calculated lipophilicity value, expressed as WLOGP, ranges from −0.4 to 5.6, with a molecular weight from 160 to 480 and a molecular refractivity from 40 to 130 and whereby the total number of atoms in the molecule is 20–70 [54]. Drug-likeness can also be characterized according to slightly different criteria adopted by Veber, which also take into account the number of rotational bonds (nROTs) and the topological value of the polar surface of the molecule (TPSA). A rotatable bond is any single bond in an acyclic system, bounded by an atom other than hydrogen. The number of rotational bonds in the molecule along with the number of hydrogen bond donors and acceptors influence the bioavailability of the substance after oral administration. To consider a compound as drug-like, the number of rotational bonds in its molecule should not exceed 10 [55]. In turn, the topological value of the molecular polar surface (TPSA) is a descriptor allowing for the assessment of the penetration of molecules through cell membranes, which is also related to the bioavailability of the compound. For substances administered orally, to ensure good absorption, the TPSA should be below 140 Å^2^ [55].

Among the designed compounds, derivatives **10a**, **10b**, **11a**, **11b**, **14a**, and **14b** do not meet Ghose’s criteria. For all these compounds, violations result from exceeding the permissible molecular weight, and, moreover, compounds **11b**, **14a**, and **14b** are characterized by molar refractions that are too high. A lipophilicity and total number of atoms in the molecule that are too high constitute an additional violation of Ghose’s criteria in the case of compounds **14a** and **14b**. These derivatives are also characterized by a TPSA value that is too high (like derivatives **7a**, **11a**, and **11b**), and, therefore, within the meaning of the principles presented by Veber, they cannot be easily absorbed drugs when administered orally.

To sum up, the weakest drug similarity is expected for compounds in which the triazole system is substituted with the most extensive groups (compounds **11a**, **11b**, **14a**, and **14b**). In the case of the designed structures, the change of the substituent in the quinoline ring from the chlorine atom in position 7 (compounds **9a** and **9b**) to the bromine atom in position 8 (compounds **10a** and **10b**) also has a significant impact.

This result does not yet rule out the possibility of using such a structure in pharmacotherapy, because alternative methods of delivering the drug substance to diseased tissues are constantly being developed (Table 4) [56].

An essential stage in the process of developing effective therapeutic agents is the initial assessment of their ADME parameters and the prediction of toxicity risks. In this study, the pkCSM software [59] was used to assess the ADMET profile of the tested compounds [60]. The calculated parameters are presented in Table 5. Among the various parameters characterizing the absorption phase, water solubility, Caco-2 permeability, absorption in the human intestine (HIA), as well as the possibility of being a p-glycoprotein (P-gp) substrate and a p-glycoprotein inhibitor were selected.

Water solubility is a factor affecting the bioavailability and absorbability of a drug when administered orally. The solubility of the tested compounds was determined using the ESOL method via the SwissADME website, which allowed for them to be assigned to specific solubility classes [61]. The obtained results indicate that in the tested group, derivatives **4a,b**–**7a,b** (triazole with an unsubstituted or substituted benzyl group) and **11a**,**b**–**13a**,**b** (with an azidothymidine fragment or a short chain substituent), for which the logS value is within range of −2.44 to −4, can be classified as soluble compounds. Compounds **8a**,**b**–**10a**,**b**, in which the substituent in the triazole ring is a methylthiophenyl group or the quinoline system, are characterized by moderate solubility (logS is in the range of −4.18 to −5.29).

The lowest solubility is expected for derivatives containing the strongly hydrophobic betulin system, i.e., the insoluble compound **14b** (logS equal to −10.08) and the slightly soluble compound **14a** (logS equal to −9.69). N-monosubstituted sulfonamides **4a**–**14a** show better solubility than their N-disubstituted analogues **4b**–**14b**, which is due to the presence of an unsubstituted hydrogen atom in the sulfonamide group, which can participate in the formation of hydrogen bonds with water molecules.

In order to enter the systemic circulation, oral drugs should have the ability to penetrate biological membranes, which may occur according to various mechanisms. The parameters determining active substances in this respect include Caco-2 permeability and absorption in the human intestine. According to the calculation model used in the pkCSM software, compounds are characterized by good permeability when the calculated value for the Caco-2 model is higher than 0.9. In the tested group, only derivatives **4a**, **4b**, **5b**, **6a**, **6b**, **8b**, and **12a** achieved this result. The calculated HIA value, ranging from 79.864 to 100%, indicates that the tested compounds will be very well absorbed by human intestines. 

Many organs that are crucial for the absorption and distribution of drugs contain p-glycoprotein, whose task is to protect normal cells against toxic substances. This transport protein also limits the absorption of drugs that are its substrates. On the other hand, in pathological cells overexpressing this protein, the use of substances that are its inhibitors to modulate P-gp activity may result in better absorption of medicinal substances used in chemotherapy [62]. Among the compounds for which the best-fit functions were obtained from docking to the Rock binding site, only compound **9b** is not a P-gp substrate. 

After administration, absorption, and entry into the circulatory system, the medicinal substance is distributed between various compartments of the body [63]. The main obstacle to drug delivery to the central nervous system (CNS) is the blood–brain barrier (BBB). Predicting the penetration of the blood–brain barrier involves determining the logBB value, which is the logarithm of the ratio of the concentration of a substance in the brain to the concentration in the plasma, at steady state [64]. All tested compounds showed negative logBB values in the range of −0.94 to 2.178, indicating that the molecules will be poorly distributed to the brain. Another parameter determining the penetration into the CNS is logPS (the logarithmic permeability surface-area product), which provides more information than logBB. PS is measured in units of mL/min/g brain and can be treated as a brain pharmacokinetic value. For compounds **6a**, **6b**, **10a**, **11a**, **11b**, **13a**, and **13b**, the logPS value is ≤−3, which classifies them as inactive in the CNS [64].

An important issue in the process of discovering new medicinal substances is predicting their metabolism. The biotransformation of drugs is associated with cytochrome P, which participates in phase I processes. For the tested compounds, interactions with the following isoforms of the cytochrome P450 monooxygenase family were determined: CYP1A2, CYP3A4, CYP2C19, and CYP2D6. Among the tested triazole derivatives, only compounds **4a**, **4b**, **5a**–**8a**, and **12b** are inhibitors of most (3 of 4) selected isoforms, which may result in poor elimination and, consequently, higher drug-induced toxicity [65].

The basic pharmacokinetic descriptors of the designed compounds are presented in Table 5.

When assessing a new chemical compound considered as a potential therapeutic entity, three basic criteria are taken into account: effectiveness, quality, and pharmacological safety. The harmful effects of new compounds on humans and the environment (animals, plants, air, and water resources) may pose a significant threat. More than 30% of drug candidates are rejected due to toxicity. The necessary preclinical tests performed in vivo in animal models require ethical approval, significant financial resources, and time [66]. Therefore, currently, in the initial toxicological assessment of new medicinal substances, in silico models are used, which are characterized by the quick implementation, data availability, and easy standardization of methods.

One of the main tools of toxicological genetics is the AMES test, which is a part of the preclinical tests detecting the mutagenic effects of new medicinal substances [67]. The in silico mutagenicity assessment (AMES) of the tested compounds **4**–**14** (**a**,**b**) showed that only derivatives **4b**–**8b**, which are tertiary amides containing a benzyl substituent at N-3 of the triazole system and secondary amide **7a**, can potentially cause negative changes in the structure of the organism’s DNA.

### 2.2. Chemistry

Sulfonamides constitute an important group of biologically active compounds. Therefore, the synthesis of sulfonamides is an important area in the field of organic synthesis. The preparation of this class of compounds typically occurs by reacting the appropriate amine with sulfonyl chlorides. It is clear that this method is effective, and its versatility has been proven; however, the use of sulfonyl chlorides causes serious storage and handling problems, as well as the generation of significant waste. Several other methods have also been described, such as the coupling of sulfonamides with aryl halides or the reaction of activated sulfonate esters with amines. However, all of these procedures require the use of volatile solvents and create at least stoichiometric amounts of undesirable by-products. Therefore, in this work, we adopted the method of obtaining alkynyl sulfonamides by the alkylation of primary sulfonamides. Because of this, the need to use solvents was eliminated, and the amount of waste was significantly reduced compared to that of the previously used method of obtaining these compounds [16] (Figure 1).

Reagents and conditions:(i)8-quinolinesulfochloride **1** (1 eq), conc. NH_4_OH, 45 °C, 0.5 h;(ii)8-quinolinesulfonamide **2** (1 eq), KOH (1.5 eq), propargyl alcohol, b.p, 6 h;(iii)8-(*N*-propargyl)quinolinesulfonamide **3a** (1 eq), CH_3_I (1.1 eq), 5% NaOH, r.t., over night;(iv)Procedure A: 8-quinolinesulfonamide **3a** or **3b** (1 eq), organic azide (1.1. eq), CuSO_4_ × 5 H_2_O, sodium acorbate, DMF/H_2_O, ambient terperatur, over night; Procedure B: organic bromide (1 eq), NaN_3_ (1.2 eq), DMF, ambient terperatur, over night and then 8-quinolinesulfonamide (1 eq), CuSO_4_ × 5 H_2_O, sodium acorbate, DMF/H_2_O, ambient terperatur, over night.

In the first step of the synthesis of the title compounds, sulfochloride **1** was treated with aqueous ammonia to obtain primary 8-quinoline sulfonamide **2** (Figure 1). This compound was subjected to an alkylation reaction with excess propargyl alcohol to obtain the secondary sulfonamide **3a**. Propargyl alcohol acted as a substrate and a solvent in this reaction, so it was used in excess. It should be mentioned that no sulfonamide dialkylation product was found in the reaction products. Part of this derivative was used as a substrate to obtain triazole derivatives of 8-quinolinesulfonamide **4a**–**14a**. The second part was subjected to another alkylation reaction with methyl iodide to sulfonamide **3b**, which was converted to derivatives **4b**–**14b**.

The title compounds were obtained from sulfonamides **3a** and **3b** using a synthetic protocol based on the copper-catalysed azide–alkyne cycloadditions (CuAAC) reaction. The catalyst necessary for the selective course of the reaction at low temperature (i.e., Cu^+^ ions) was produced in the reaction medium by reducing Cu^2+^ ions with sodium ascorbate. In most cases, the appropriate azides were used as the second substrate in the CuAAC reaction (Procedure A). The exception was aliphatic azides: 1-azobutane and ethyl-3-azidopropanoate. These azides, because of the high nitrogen content in the molecule, may be unstable. Therefore, they were prepared in the reaction medium from the appropriate aliphatic bromide and sodium azide (Procedure B). The reaction was carried out in a DMF solution, and the products were isolated by pouring the reaction mixture into water and filtering. ^1^H NMR, ^13^C NMR and HR MS spectra are presented in the Appendix A.

### 2.3. In Vitro Studies

Hybrids **4**–**14** were tested as anticancer agents against four human cancer cell lines: colon cancer (Caco-2), glioblastoma (SNB-19), lung cancer (A549), ovarian cancer (SKOV-3), and normal human dermal fibroblasts (NHDFs). Cisplatin was used as the reference substance. The results are presented as the compound concentration of the compound (μM) that inhibits the growth of 50% of cancer cells compared to control cells (IC_50_). A negative value (Neg) means that the compound concentration of the tested hybrids was greater than 500 µM. As can be seen in Table 6, the derivatives **4**–**14** show high activity against colon cancer (Caco-2) and glioma (SNB-19). The highest activity against this line (≥0.3 µM) is demonstrated by compounds **9b** and **10b**, for which the IC_50_ value is comparable with the value determined for cisplatin. The most active derivatives against glioma (≤0.4 µM) are compounds **8b**, **9b**, **10b**, and **13a**. The structure–activity relationship indicates that, in general, tertiary sulfonamides are more active in relation to these two tested lines. An analysis of the **9b** derivative complex with the target protein obtained as a result of molecular docking (Figure 5) indicated that the methyl group present at the sulfonamide nitrogen atom does not significantly affect the stability of the complex. It is generated only by a weak van der Waals interaction with Arg84. This may suggest that the increased activity of tertiary sulfonamides may be caused, for example, by the greater lipophilicity of these compounds compared to analogous secondary sulfonamides, which may affect the transport of the tested substances through biological membranes. There is also a visible difference in activity depending on the type of group present on the triazole ring (substituent C in Figure 2). The results obtained indicate that hybrids containing aromatic systems as a C substituent in their structure, especially another quinoline molecule (compounds **9** and **10**), have high activity.

An analysis of the activity of derivatives **4**–**14**, relative to the A549 cell line, indicated their medium or low activity. Only the activities of tertiary sulfonamides **5b** and **9b** are in the nanomolar concentration range. The activities of the title compounds toward the SKOV-3 line also reach low values. The compound with the highest activity, also in this case, is derivative **9b**.

The in vitro tests performed allowed for the selection of four compounds with a wide spectrum of activity from among 22 hybrids. These are derivatives **5b**, **9b**, **10b**, and **11b**. Three of them, i.e., compounds **5b**, **10b,** and **11b**, have activity against normal cells. However, derivative **9b** has no activity towards NHDF. These results are consistent with the in silico experiment, as bundle **9b** showed the highest inhibitory potential against the target protein in these studies.

## 3. Materials and Methods

### 3.1. In Silico

The appropriate three-dimensional structures of the docking ligands were generated using Gaussian 16 (Revision C-01). The lowest energy conformations were obtained by performing optimisation calculations using the basis (DFT and B3LYP) method and 6–311 + G (d, p). The crystallographic structure of the protein, along with the native ligand (fasudil), was obtained from the Protein Data Bank (https://www.rcsb.org/) (accessed on 27 September 2023). We used the three-dimensional crystal structure of ROCK1 with PDB code 2ESM. Docking was performed with AutoDock Vina version 1.12 [68], supported by PyRx version 0.8 [69]. The volume of the region of interest used for docking was defined as 25 × 25 × 25 Å, with a centre point at coordinates X = 51.5, Y = 99.9, and Z = 28.5 Å. The AutoDock Vina programme generates 9 complexes for each ligand. Only the lowest-energy complexes were selected for further analysis. All results obtained are presented in kcal/mol. The molecular coupling results of the title compounds were visualised using the BIOVIA Discovery Studio version 19.1.0. 18287 programme [70].

The lowest-energy complex of ligand **9b** with the target protein generated by AutoDock Vina was selected for molecular dynamics calculations. Trajectory calculations were performed in Nanoscale Molecular Dynamics ver. 2.13 (NAMD) [71]. The input files for the calculations were generated using the VMD version 1.9 programme [72]. The ligand parameters for the force field were obtained from the CGenFF server [73]. Parameterized ligands were introduced into the protein and saved as a protein–ligand complex using the QwikMD version 1.9.3 software [74]. The protein–ligand complex was placed in the centre of the box and solvated with water molecules using the TIP3P water box. The electric charges were neutralised by adding Na+ and Cl^−^ ions at a concentration of 0.15 M. CHARMM 36 (Chemistry at HARvard Macromolecular Mechanics) was used to parameterize the protein. To minimize and equilibrate the complexes in the water cell, we adopted the force-field parameters excluding a scaling of 1.0. All atoms, including hydrogen atoms, are explicitly illustrated. The initial energy was minimised for 2000 steps at a constant temperature (310 K), and an additional 144,000 steps were then simulated using Langevin dynamics to control the kinetic energy, temperature, and/or pressure of the system. The protein–ligand complex was equilibrated using 500,000 minimisation steps and 25,000,000 runs for 50 ns. The resulting trajectory file obtained using NAMD was analysed and visualized using VMD.

Physicochemical properties and drug-likeness parameters were obtained using SwissADME, the free web tool available online (https://www.swissadme.ch) (accessed on 27 September 2023). Selected ADME parameters were calculated from the pkCSM machine-learning platform using distance–pharmacophore patterns encoded as graph-based signatures (https://biosig.lab.uq.edu.au/pkcsm/) (accessed on 27 September 2023). 

### 3.2. Chemistry

#### 3.2.1. General Chemistry Methods 

Reagents used in this research were purchased from Sigma-Aldrich, Fluorochem and AlfaAesar.

NMR spectra (^1^H, ^13^C, HSQC, and HMBC) were recorded on a Bruker Fourier 300 (Bruker Corporation, Billerica, MA, USA) in a CDCl_3_ or DMSO-d6 solution and calibrated to residual solvent signals. The values of the coupling constants are given in Hertz (Hz), and the resulting resonance peaks are described as follows: br s. (broad singlet), br d. broad doublet), s (singlet), d (doublet), t (triplet), q (quartet), dd (doublet of doublets), dt (double of triplets), td (triplet of doublets), tt (triplet of triplets), ddd (doublet of doublet of doublets), dq (doublet of quartets), and m (multiplet). The numbering system shown in Figure 9 was used to describe the NMR spectra.

#### 3.2.2. Synthesis of Quinolinesulfonamide **2**

A total of 0.57 g (2.5 mmol) of 8-Quinolinesulfonyl chloride (**1**) and conc. NH_4_OH (12.5 mL) were stirred at 45 °C for 0.5 h. An excess of ammonia was evaporated under vacuum. Then, water was added up to a volume of 10 mL. The solid was filtered off and washed with cold water. It was finally recrystallised from 10 % aqueous EtOH.

#### 3.2.3. Synthesis of Quinolinesulfonamide **3a**

A total of 0.221 g (1 mmol) of 8-Quinolinosulfonamide (**2**), 0.086 g (1.5 mmol) of KOH, and 1 mL (0.948 g, 16.9 mmol) of propargyl alcohol were stirred under reflux for 10 h. The excess alcohol was distilled off under a reduced pressure. A total of 15 mL of water was added to the residue, filtered, and air-dried. The product was purified by column chromatography with a silica gel using ethyl acetate as an eluent.

#### 3.2.4. Synthesis of Quinolinesulfonamide **3b**

A total of 0.492 g (2 mmol) of quinolinesulfonamide (**3a**) was dissolved in 2 mL of a 5% aqueous KOH solution. Then, 0.136 mL of methyl iodide (0.312 g, 2.2 mmol) was added dropwise to the obtained solution with vigorous stirring. The mixture was stirred at room temperature overnight. The obtained product was filtered off, washed on the filter with 1 mL of water, and air-dried. The product was purified by column chromatography with a silica gel using ethyl acetate as an eluent.

The physical and spectroscopic properties of 8*-N*-(prop-2-ynyl)quinolinesulfonamide **(3a**) and 8-*N*-methyl-*N*-(prop-2-ynyl)quinolinesulfonamide (**3b**) were consistent with the data from the literature [20].

#### 3.2.5. Synthesis of Triazoles **4**–**14**

Procedure A:

A solution of 0.02 g (0.10 mmol) of sodium ascorbate in 0.5 mL of water and a solution of 0.0125 g (0.05 mmol) of CuSO_4_ × 5H_2_O in 0.5 mL of water were prepared. Both aqueous solutions were mixed, and the resulting suspension was immediately added to a solution containing 0.5 mmol of sulfonamide **3a** or **3b** in 5 mL of DMF and 0.75 mmol of the appropriate azide. The reaction mixture was stirred overnight at room temperature. In order to isolate the reaction products, the reaction mixture was poured into 50 mL of water and filtered. Products **4**–**11** and **14** were purified by SiO_2_ column chromatography, and ethyl acetate was used as the mobile phase.

Procedure B:

A solution containing 0.036 g (0.55 mmol) of sodium azide and 0.5 mmol of 1-bromobutane or ethyl-3-bromopropanoate in 5 mL of DMF was stirred overnight at room temperature. Then, the appropriate sulfonamide **3a** or **3b** (0.5 mmol) and an aqueous solution of Cu + salt obtained from sodium ascorbate and copper(II) sulphate(IV) pentahydrate were added, as described in procedure A. The mixture was stirred overnight at room temperature, then poured into 50 mL of water and filtered. Products **12** and **13** were purified by column chromatography with SiO_2_, and ethyl acetate was used as the mobile phase.

8*-N*-[(1-Benzyl-1*H*-1,2,3-triazol-4-yl)methyl]quinolinesulfonamide (**4a**):

Yield, 85 %; m.p., 125–126 °C. ^1^H NMR (DMSO-d_6_ and 300 MHz) δ, 4.19 (d, J = 6.0 Hz, 2H, H-10); 5.37 (s, 2H, H-13); 7.15–7.17 (m, 2H, H-15, and H-15′); 7.32–7.29 (m, 3H, H-16, H-16′, and H-17); 7.66–7.71 (m, 4H, H-3, H-6, H-9, and H-12); 8.23–8.28 (m, 2H, H-5, and H-7); 8.50 (dd, J = 8.1 Hz, J = 1.5 Hz, 1H, and H-4); 9.02 (dd, J = 4.2 Hz, J = 1.5 Hz, 1H, and H-2). ^13^C NMR (DMSO-d_6_, 75 MHz) δ: 39.0 (C-10), 53.0 (C-13), 122.9 (C-3), 123.6 (C-12), 126.1 (C-6), 128.4 (C-15 and C-15′), 128.6 (C-17), 128.8 (C-4a), 129.2 (C-16 and C-16′), 130.9 (C-7), 134.0 (C-5), 136.3 (C-14), 137.0 (C-8), 137.4 (C-4), 143.0 (C-8a), 144.2 (C-11), 151.7 (C-2). HRMS (ESI) *m*/*z*: C_19_H_18_N_5_O_2_S [M + H]^+^; calcd., 380.1181; found, 380.1182.

8-*N*-Methyl-*N*-[(1-benzyl-1*H*-1,2,3-triazol-4-yl)methyl]quinolinesulfonamide (**4b**):

Yield, 94 %; m.p., 103–105 °C. ^1^H NMR (DMSO-d_6_ and 300 MHz) δ, 2.81 (s, 3H, and H-9); 4.63 (s, 2H and H-10); 5.53 (s, 2H, and H-13); 7.25–7.27 (m, 2H, H-15, and H-15′); 7.33–7.40 (m, 3H-16, H-16′, and H-17); 7.66–7.77 (m, 2H, H-3, and H-6); 7.98 (s, 1H, and H-12); 8.29 (dd, J = 8.4 Hz, J = 1.2 Hz, 1H, and H-5); 8.38 (dd, J = 7.5 Hz, J = 1.2 Hz, 1H, and H-7); 8.52 (dd, J = 8.1 Hz, J = 1.5 Hz, 1H, and H-4); 9.06 (dd, J = 4.2 Hz, J = 1.5 Hz, 1H, and H-2). ^13^C NMR (DMSO-d_6_, 75 MHz) δ: 35.4 (C-9), 46.1 (C-10), 53.2 (C-13), 122.9 (C-3), 124.1 (C-12), 126.2 (C-6), 128.4 (C-15 and C-15′), 128.6 (C-17), 129.1 (C-4a), 129.2 (C-16 and C-16′), 133.1 (C-7), 134.4 (C-5), 136.5 (C-8), 136.9 (C-14), 137.3 (C-4), 143.7 (C-8a), 144.0 (C-11), 151.8 (C-2). HRMS (ESI) *m*/*z*: C_20_H_19_N_5_NaO_2_S [M + Na]^+^; calcd., 416.1157; found, 416.1162.

8-*N*-{[1-(4-Cyanobenzyl)-1*H*-1,2,3-triazol-4-yl]methyl}quinolinesulfonamide (**5a**):

Yield, 87 %; m.p., 110–111 °C. ^1^H NMR (DMSO-d_6_ and 300 MHz) δ, 4.20 (d, J = 6.0 Hz, 2H, and H-10); 5.51 (s, 1H, and H-13); 7.29 (d, J = 8.4 Hz, 2H, H-15, and H-15′); 7.66–7.73 (m, 3H, H-3, H-6, and H-9); 7.78 (s, 2H, and H-12); 7.84 (d, J = 8.4 Hz, 2H, H-16, and H-16′); 8.25 (m, 2H, H-5, and H-7); 8.50 (dd, J = 8.1 Hz, J = 1.5 Hz, 1H, and H-4); 9.02 (dd, J = 4.2 Hz, J = 1.5 Hz, 1H, and H-2). ^13^C NMR (DMSO-d_6_ and 75 MHz) δ: 38.9 (C-10), 52.3 (C-13), 111.3 (C-17), 119.0 (C-18), 122.9 (C-3), 124.1 (C-12), 126.1 (C-6), 128.8 (C-4a), 129.1 (C-15 and C-15′), 130.9 (C-7), 133.14 (C-16 and C-16′), 134.0 (C-5), 137.0 (C-8), 137.4 (C-4), 141.8 (C-14), 143.0 (C-8a), 144.4 (C-11), 151.7 (C-2). HRMS (ESI) *m*/*z*: C_20_H_17_N_6_O_2_S [M + H]^+^; calcd., 405.1134; found, 405.1137.

8-*N*-Methyl-*N*-{[1-(4-cyanobenzyl)-1*H*-1,2,3-triazol-4-yl]methyl}quinolinesulfonamide (**5b**):

Yield, 92 %; m.p., 112–113 °C. ^1^H NMR (DMSO-d_6_ and 300 MHz) δ, 2.82 (s, 3H, and H-10); 4.65 (s, Hz, 2H, and H-10); 5.66 (s, 2H, and H-13); 7.39 (d, J = 8.4 Hz, 2H, H-15, and H-15′); 7.66–7.78 (m, 2H, H-3, and H-6); 7.85 (d, J = 8.4 Hz, 2H, H-16, and H-16′); 8.06 (s, 2H, and H-12); 7.84 (d, J = 8.4 Hz, 2H, H-16, and H-16′); 8.29 (dd, J = 8.4 Hz, J = 1.2 Hz, 1H, and H-5); 8.38 (dd, J = 7.5 Hz, J = 1.2 Hz, 1H, and H-7); 8.52 (dd, J = 8.1 Hz, J = 1.5 Hz, 1H, and H-4); 9.06 (dd, J = 4.2 Hz, J = 1.5 Hz, 1H, and H-2). ^13^C NMR (DMSO-d_6_ and 75 MHz) δ: 35.4 (C-9); 46.1 (C-10); 52.5 (C-13); 111.4 (C-17); 119.0 (C-18); 122.9 (C-3); 124.5 (C-12); 126.2 (C-6); 129.1 (C-15, C-15′, and C-4a); 133.1 (C-7); 133.2 (C-16 and C-16′); 134.4 (C-5); 136.8 (C-8); 137.3 (C-4); 141.9 (C-14); 143.7 (C-8a); 144.2 (C-11); 151.8 (C-2). HRMS (ESI) *m*/*z*: C_21_H_19_N_6_O_2_S [M + H]^+^; calcd., 419.1290; found, 419.1290.

8-*N*-{[1-(4-Fluorobenzyl)-1*H*-1,2,3-triazol-4-yl]methyl}quinolinesulfonamide (**6a**):

Yield, 91 %; m.p., 118–119 °C. ^1^H NMR (DMSO-d_6_ and 300 MHz) δ, 4.18 (d, J = 6.0 Hz, 2H, and H-10); 5.36 (s, 2H, and H-13); 7.16–7.26 (m, 4H, H-15, H-15′, H-16, and H-16′); 7.66–7.72 (m, 4H, H-3, H-6, H-9, and H-12); 8.23–8.27 (m, 2H, H-5, and H-7); 8.50 (dd, J = 8.4 Hz, J = 1.5 Hz, 1H, and H-4); 9.01 (dd, J = 4.2 Hz, J = 1.5 Hz, 1H, and H-2). ^13^C NMR (DMSO-d_6_ and 75 MHz) δ: 39.0 (C-10); 52.2 (C-13); 116 (d, ^2^J_C-F_ = 21 Hz, C-16, and C-16′); C-122.9 (C-3); 123.5 (C-12); 126.1 (C-6); 128.8 (C-4a);, 130.7 (d, ^3^J_C-F_ = 9.0 Hz, C-15, and C-15′); 130.9 (C-7); 132.5 (d, ^4^J_C-F_ = 3.0 Hz, and C-14); 134.0 (C-5); 137.0 (C-4); 137.4 (C-8); 143.0 (C-8a); 144.3 (C-11); 151.7 (C-2); 162.3 (d, ^1^J_C-F_ = 162.3 Hz, and C-17). HRMS (ESI) *m*/*z*: C_19_H_17_FN_5_O_2_S [M + H]^+^; calcd., 398.1087; found, 398.1080.

8-*N*-Methyl-*N*-{[1-(4-fluorobenzyl)-1*H*-1,2,3-triazol-4-yl]methyl}quinolinesulfonamide (**6b**):

Yield: 90 %; m.p., 85–86 °C. ^1^H NMR (DMSO-d_6_ and 300 MHz) δ: 2.81 (s, 3H, and H-9); 4.63 (s, 2H, and H-10); 5.52 (s, 2H, and H-13); 7.18–7.24 (m, 2H, H-15, and H-15′); 7.32–7.37 (m, 2H, H-16, and H-16′); 7.66-7.77 (m, 2H, H-3, and H-6); 7.99 (s, 1H, and H-12); 8.29 (dd, J = 8.4 Hz, J = 1.2 Hz, 1H, and H-5); 8.38 (dd, J = 7.5 Hz, J = 1.2 Hz, 1H, and H-7); 8.52 (dd, J = 8.4 Hz, J = 1.5 Hz, 1H, and H-4); 9.05 (dd, J = 4.2 Hz, J = 1.5 Hz, 1H, and H-2). ^13^C NMR (DMSO-d_6_ and 75 MHz) δ: 35.4 (C-9); 46.1 (C-10); 52.4 (C-13); 116.1 (d, ^2^J_C-F_ = 21.8 Hz, C-16, and C-16′); C-122.9 (C-3); 124.0 (C-12); 126.2 (C-6); 129.1 (C-4a); 130.7 (d, ^3^J_C-F_ = 8.3 Hz, C-15, and C-15′); 132.6 (d, ^4^J_C-F_ = 3.0 Hz, and C-14); 133.1 (C-7); 134.4 (C-5); 136.9 (C-8); 137.3 (C-4); 143.7 (C-8a); 144.0 (C-11); 151.8 (C-2); 162.3 (d, ^1^J_C-F_ = 243.8 Hz, and C-17. HRMS (ESI) *m*/*z*: C_20_H_19_FN_5_O_2_S [M + H]^+^; calcd., 412.1243; found, 412.1243.

8-*N*-{[1-(4-Nitrobenzyl)-1*H*-1,2,3-triazol-4-yl]methyl}quinolinesulfonamide (**7a**):

Yield, 94 %; m.p., 135–136 °C. ^1^H NMR (DMSO-d_6_ and 300 MHz) δ, 4.20 (d, J = 6.0 Hz, 2H, and H-10); 5.57 (s, 2H, and H-13); 7.39 (d, J = 8.4 Hz, 2H, H-15, and H-15′); 7.65–7.73 (m, 3H, H-3, H-6, and H-9); 7.82 (s, 1H, and H-12); 8.20–8.31 (m, 4H, H-5, H-7, H-16, and H-16′); 8.49 (dd, J = 8.1 Hz, J = 1.5 Hz, 1H, and H-4); 9.02 (dd, J = 4.2 Hz, J = 1.5 Hz, 1H, and H-2). ^13^C NMR (DMSO-d_6_ and 75 MHz) δ: 38.9 (C-10), 52.0 (C-13), 122.9 (C-3), 124.1 (C-12), 124.3 (C-16 and C-16′), 126.1 (C-6), 128.8 (C-4a), 129.4 (C-15 and C-15′), 131.0 (C-7), 134.0 (C-5), 137.0 (C-8), 137.4 (C-4), 143.0 (C-8a), 143 (C-14), 144.5 (C-11), 147.7 (C-17), 151.7 (C-2). HRMS (ESI) *m*/*z*: C_19_H_17_N_6_O_4_S [M + H]^+^; calcd., 425.1032; found, 425.1038.

8-*N*-Methyl-*N*-{[1-(4-nitrobenzyl)-1*H*-1,2,3-triazol-4-yl]methyl}quinolinesulfonamide (**7b**):

Yield, 93 %; m.p., 103–105 °C. ^1^H NMR (DMSO-d_6_ and 300 MHz) δ, 2.81 (s, 3H, and H-9); 4.66 (s, 2H, and H-10); 5.73 (s, 2H, and H-13); 7.49 (d, J = 8.4 Hz, 2H, H-15, and H-15′); 7.66–7.78 (m, 2H, H-3, and H-6); 8.09 (s, 1H, and H-12); 8.23–8.31 (m, 3H, H-5, H-16, and H-16′); 8.38 (dd, J = 7.5 Hz, J = 1.2 Hz, 1H, and H-7); 8.52 (dd, J = 8.1 Hz, J = 1.5 Hz, 1H, and H-4); 9.06 (dd, J = 4.2 Hz, J = 1.5 Hz, 1H, and H-2). ^13^C NMR (DMSO-d_6_ and 75 MHz) δ: 35.4 (C-9), 46.1 (C-10), 52.3 (C-13), 122.9 (C-3), 124.4 (C-16 and C-16′), 124.6 (C-12), 126.2 (C-6), 129.1 (C-4a), 129.5 (C-15 and C-15′), 133.1 (C-7), 134.4 (C-5), 136.8 (C-8), 137.3 (C-4), 143.7 (C-8a), 143.9 (C-14), 144.2 (C-11), 147.8 (C-17), 151.8 (C-2). HRMS (ESI) *m*/*z*: C_20_H_19_N_6_O_4_S [M + H]^+^; calcd., 439.1188; found, 439.1182.

8-*N*-[(1-Methylthiophenyl-1*H*-1,2,3-triazol-4-yl)methyl]quinolinesulfonamide (**8a**):

Yield, 89 %; m.p., 106–107 °C. ^1^H NMR (DMSO-d_6_ and 300 MHz) δ, 4.16 (d, J = 6.0 Hz, 2H, and H-10); 5.74 (s, 2H, and H-13); 7.28–7.33 (m, 5H, H-15, H-15′, H-16, H-16′, and H-17); 7.66–7.72 (m, 4H, H-3, H-6, H-9, and H-12); 8.21–8.28 (m, 2H, H-5, and H-7); 8.50 (dd, J = 8.1 Hz, J = 1.5 Hz, 1H, and H-4); 9.02 (dd, J = 4.2 Hz, J = 1.5 Hz, 1H, and H-2). ^13^C NMR (DMSO-d_6_ and 75 MHz) δ: 38.9 (C-10), 51.8 (C-13), 122.9 (C-3), 123.2 (C-12), 126.1 (C-6), 128.0 (C-17), 128.8 (C-4a), 129.7 (C-16 and C-16′), 130.6 (C-15 and C-15′), 131.0 (C-7), 133.1 (C-14), 134.0 (C-5), 137.0 (C-8), 137.4 (C-4), 143.0 (C-8a), 144.6 (C-11), 151.7 (C-2). HRMS (ESI) *m*/*z*: C_19_H_18_N_5_O_2_S_2_ [M + H]^+^; calcd., 412.0902; found, 412.0901.

8-*N*-Methyl-*N*-[(1-methylthiophenyl-1*H*-1,2,3-triazol-4-yl)methyl]quinolinesulfonamide (**8b**):

Yield, 90 %; m.p., 118–119 °C. ^1^H NMR (DMSO-d_6_ and 300 MHz) δ, 2.74 (s, 3H, and H-9); 4.61 (s, 2H, and H-10); 5.89 (s, 2H, and H-13); 7.29–7.39 (m, 5H, H-15, H-15′, H-16, H-16′, and H-17); 7.66–7.78 (m, 2H, H-3, and H-6); 7.87 (s, 1H, and H-12); 8.30 (dd, J = 8.4 Hz, J = 1.2 Hz, 1H, and H-5); 8.38 (dd, J = 7.5 Hz, J = 1.2 Hz, 1H, and H-7); 8.52 (dd, J = 8.1 Hz, J = 1.5 Hz, 1H, and H-4); 9.06 (dd, J = 4.2 Hz, J = 1.5 Hz, 1H, and H-2). ^13^C NMR (DMSO-d_6_ and 75 MHz) δ: 35.2 (C-9), 46.0 (C-10), 52.1 (C-13), 122.9 (C-3), 123.8 (C-12), 126.2 (C-6), 128.2 (C-17), 129.2 (C-4a), 129.7 (C-16 and C-16′), 131.2 (C-15 and C-15′), 132.8 (C-14), 133.1 (C-7), 134.5 (C-5), 136.8 (C-8), 137.4 (C-4), 143.7 (C-8a), 144.2 (C-11), 151.9 (C-2). HRMS (ESI) *m*/*z*: C_20_H_20_N_5_O_2_S_2_ [M + H]^+^; calcd, 426.1058; found, 426.1048.

The physical and spectroscopic properties of 8-*N*-{[1-(7-chloroquinolin-4-yl)-1*H*-1,2,3-triazol-4-yl]methyl}quinolinesulfonamide (**9a**) were consistent with literature data. [20].

8-*N*-Methyl-*N*-{[1-(7-chloroquinolin-4-yl)-1*H*-1,2,3-triazol-4-yl]methyl}quinolinesulfonamide (**9b**):

Yield, 83 %; m.p., 189–190 °C. ^1^H NMR (DMSO-d_6_ and 300 MHz) δ, 2.98 (s, 3H, and H-9); 4.83 (s, 2H, and H-10); 7.67–7.89 (m, 5H, H-3, H-3′, H-5′, H-6, and H-6′); 8.28–8.32 (m, 2H, H-5, and H-8′); 8.41 (dd, J = 8.4 Hz, J = 1.5 Hz, 1H, and H-7); 8.50 (dd, J = 8.1 Hz, J = 1.5 Hz, 1H, and H-4); 8.65 (s, 1H, and H-12); 9.10 (dd, J = 4.2 Hz, J = 1.5 Hz, 1H, and H-2); 9.13 (d, J = 4.8 Hz, 1H, and H-2′). ^13^C NMR (DMSO-d_6_ and 75 MHz) δ, 35.9 (C-9), 45.8 (C-10), 117.4 (C-3′), 120.6 (C-4a’), 123.0 (C-3), 125.9 (C-5′), 126.2 (C-6 and C-12), 128.6 (C-8′), 129.1 (C-4a), 129.4 (C-6′), 133.3 (C-7), 134.5 (C-5), 135.8 (C-7′), 136.9 (C-8), 137.3 (C-4), 140.7 (C-4′), 143.7 (C-8a), 144.4 (C-11), 149.8 (C-8a’), 151.8 (C-2), 152.8 (C-2′). HRMS (ESI) *m*/*z*: C_22_H_18_ClN_6_O_2_S [M + H]^+^; calcd., 465.0900; found, 465.0896.

8-*N*-{[1-(8-Bromoquinolin-4-yl)-1*H*-1,2,3-triazol-4-yl]methyl}quinolinesulfonamide (**10a**):

Yield, 91 %; m.p., 194–195 °C. ^1^H NMR (DMSO-d_6_ and 300 MHz) δ, 4.40 (d, J = 6.0 Hz, 2H, and H-10); 7.49 (dd, J = 8.1 Hz, J = 1.2 Hz, 1H, and H-7′); 7.579–7.73 (m, 4H, H-3, H-3′, H-6, and H-6′); 7.84 (t, J = 6.0 Hz, 1H, and H-9); 8.21 (dd, J = 8.2 Hz, J = 1.2 Hz, 1H, and H-5′); 8.29–8.32 (m, 3H, H-5, H-7, and H-12); 8.45 (dd, J = 8.1 Hz, J = 1.5 Hz, 1H, and H-4); 9.05 (dd, J = 4.2 Hz, J = 1.5 Hz, 1H, and H-2); 9.17 (d, J = 1.5 Hz, 1H, and H-2′). ^13^C NMR (DMSO-d_6_ and 75 MHz) δ: 38.7 (C-10), 118.0 (C-3′), 122.9 (C-3), 123.4 (C-8′), 123.6 (C-7′), 125.2 (C-4a’), 126.1 (C-12), 126.2 (C-6), 128.7 (C-4a), 129.4 (C-6′), 131.0 (C-7), 134.1 (C-5′), 134.8 (C-5), 137.1 (C-8), 137.4 (C-4), 140.8 (C-4′), 143.0 (C-8a), 144.4 (C-11), 145.9 (C-8a’), 151.7 (C-2), 152.2 (C-2′). IR (KBr, cm^−1^) ν_max_: 3302 (≡C-H), 3041 and 2974 (CH_2_ and CH_3_), 2129 (C≡C), 1340 (S=O), 1155 (S=O). HRMS (ESI) *m*/*z*: C_21_H_16_BrN_6_O_2_S [M + H]^+^; calcd., 495.0239; found, 495.0239.

8-*N*-Methyl-*N*-{[1-(8-bromoquinolin-4-yl)-1*H*-1,2,3-triazol-4-yl]methyl}quinolinesulfonamide (**10b**):

Yield, 87 %; m.p., 206–207 °C. ^1^H NMR (DMSO-d_6_ and 300 MHz) δ, 2.98 (s, 3H, and H-9); 4.84 (s, 2H, and H-10); 7.61–7.83 (m, 5H, H-3, H-3′, H-6, H-6′, and H-7′); 8.31 (m, 2H, H-5, and H-5′); 8.44 (dd, J = 7.5 Hz, J = 1.2 Hz, 1H, and H-7); 8.50 (dd, J = 8.1 Hz, J = 1.5 Hz, 1H, and H-4); 8.63 (s, 1H, and H-12); 9.10 (dd, J = 4.2 Hz, J = 1.5 Hz, 1H, and H-2); 9.21 (d, J = 1.5 Hz, 1H, and H-2′). ^13^C NMR (DMSO-d_6_ and 75 MHz) δ: 35.8 (C-9), 45.8 (C-10), 118.4 (C-3′), 123.0 (C-3), 123.7 (C-7′ and C-8′), 125.2 (C-4a’), 126.3 (C-6), 126.5 (C-12), 129.1 (C-4a), 129.5 (C-6′), 131.3 (C-7), 134.5 (C-5), 134.8 (C-5′), 136.9 (C-8), 137.4 (C-4), 141.1 (C-4′), 143.7 (C-8a), 144.3 (C-11), 146.0 (C-8a′), 151.8 (C-2), 152.3 (C-2′). HRMS (ESI) *m*/*z*: C_22_H_18_BrN_6_O_2_S [M + H]^+^; calcd., 509.0395; found, 509.0388.

8-*N*-({1-[(2*R*, 3*S*, 5*S*)-2-(Hydroxymethyl)-5-(5-methyl-2,4-dioxo-3,4-dihydropyrimidin-1(2*H*)-yl)tertrahydrofuran-3-yl]-1*H*-1,2,3-triazol-4-yl}methyl)quinolinesulfonamide (**11a**):

Yield, 93 %; m.p., 149–150 °C. ^1^H NMR (DMSO-d_6_ and 300 MHz) δ, 1.81 (s, 3H, and H-24); 2.49 (m, 2H, and H-14); 2.44–3.64 (m, 2H, and H-17); 3.64–3.88 (m, 1H, and H-16); 4.21 (d, J = 6.0 Hz, 2H, and H-10); 5.08–5.14 (m, 1H, and H-13); 5.25 (t, J = 5.1 Hz, 1H, and H-18); 6.30 (t, J = 6.6 Hz, 1H, and H-15); 7.65–7.77 (m, 4H, H-3, H-6, H-9, and H-23); 7.83 (s, 1H, and H-12); 8.23–7.27 (m, 2H, H-5, and H-7); 8.50 (dd, J = 8.1 Hz, J = 1.5 Hz, 1H, and H-4); 9.04 (dd, J = 4.2 Hz, J = 1.5 Hz, 1H, and H-2); 11.37 (s, 1H, and H-20). ^13^C NMR (DMSO-d_6_ and 75 MHz) δ: 12.7 (C-24), 37.4 (C-14), 59.5 (C-13), 61.2 (C-17), 84.3 (C-16), 84.8 (C-15), 110.1 (C-22), 122.9 (C-3), 123.1 (C-12), 126.1 (C-6), 128.7 (C-4a), 130.9 (C-7), 134.0 (C-5), 136.7 (C-23), 137.0 (C-8), 137.4 (C-4), 143.0 (C-8a), 143.9 (C-11), 150.7 (C-19), 151.7 (C-2). 164.2 (C-21). HRMS (ESI) *m*/*z*: C_22_H_24_N_7_O_6_S [M + H]^+^; calcd., 514.1509; found, 514.1499.

8-*N*-Methyl-*N*-({1-[(2*R*, 3*S*, 5*S*)-2-(hydroxymethyl)-5-(5-methyl-2,4-dioxo-3,4-dihydropyrimidin-1(2*H*)-yl)tertrahydrofuran-3-yl]-1*H*-1,2,3-triazol-4-yl}methyl)quinolinesulfonamide (**11b**):

Yield, 91 %; m.p., 176–178 °C. ^1^H NMR (DMSO-d_6_ and 300 MHz) δ, 1.81 (s, 3H, and H-24); 2.61 (m, 2H, and H-14); 2.84 (s, 3H, and H-9); 3.65 (m, 2H, and H-17); 4.03–4.11 (m, 1H, and H-16); 4.65 (s, 2H, and H-10); 5.27–5.33 (m, 2H, H-13, and H-18); 6.40 (t, J = 6.6 Hz, 1H, and H-15); 7.68–7.81 (m, 3H, H-3, H-6, and H-23); 8.14 (s, 1H, and H-12); 8.30 (dd, J = 8.4 Hz, J = 1.2 Hz, 1H, and H-5); 8.38 (dd, J = 7.5 Hz, J = 1.2 Hz, 1H, and H-7); 8.52 (dd, J = 8.1 Hz, J = 1.5 Hz, 1H, and H-4); 9.07 (dd, J = 4.2 Hz, J = 1.5 Hz, 1H, and H-2); 11.38 (s, 1H, and H-20). ^13^C NMR (DMSO-d_6_ and 150MHz) δ: 12.7 (C-24), 35.6 (C-9), 37.5 (C-14), 59.6 (C-13), 61.2 (C-17), 84.3 (C-16), 84.9 (C-15), 110.1 (C-22), 122.9 (C-3), 123.6 (C-12), 126.3 (C-6), 129.1 (C-4a), 133.1 (C-7), 134.4 (C-5), 136.7 (C-23), 136.9 (C-8), 137.3 (C-4), 143.7 (C-8a), 143.9 (C-11), 150.9 (C-19), 151.8 (C-2). 164.2 (C-21). HRMS (ESI) *m*/*z*: C_23_H_26_N_7_O_6_S [M + H]^+^; calcd., 528.1665; found, 528.1659.

The physical and spectroscopic properties of 8-*N*-{[1-(1-butyl)-1*H*-1,2,3-triazol-4-yl]methyl}quinolinesulfonamide (**12a**) were consistent with literature data [20].

8-*N*-Methyl-*N*-{[1-(1-butyl)-1*H*-1,2,3-triazol-4-yl]methyl}quinolinesulfonamide (**12b**):

Yield, 94 %; m.p., 74-75 ^o^C. ^1^H NMR (DMSO-d_6_, 300 MHz) δ, 0.87 (t, J = 7.2 Hz, 3H, and H-16); 1.16 (sext, J = 7.2 Hz, 2H, and H-15); 1.67 (quint, J = 7.2 Hz, 2H, and H-14); 2.83 (s, 3H, and H-9); 4.26 (t, J = 7.2 Hz, 2H, and H-13); 4.62 (s, 2H, and H-10); 7.67–7.79 (m, 2H, H-3, and H-6); 7.87 (s, 1H, and H-12); 8.30 (dd, J = 8.1 Hz, J = 1.2 Hz, 1H, and H-5); 8.39 (dd, J = 7.5 Hz, J = 1.2 Hz, 1H, and H-7); 8.54 (dd, J = 8.1 Hz, J = 1.8 Hz, 1H, and H-4); 9.07 (dd, J = 4.2 Hz, J = 1.8 Hz, 1H, and H-2). ^13^C NMR (DMSO-d_6_ and 75 MHz) δ: 13.7 (C-16), 19.5 (C-15), 32.1 (C-14), 35.4 (C-9), 46.0 (C-10), 49.4 (C-13), 122.9 (C-3), 123.7 (C-12), 126.2 (C-6), 129.1 (C-4a), 133.1 (C-7), 134.4 (C-5), 137.0 (C-8), 137.3 (C-4), 143.4 (C-11), 143.7 (C-8a), 151.8 (C-2). HRMS (ESI) *m*/*z*: C_17_H_22_N_5_O_2_S [M + H]^+^; calcd., 360.1494; found, 360.1491.

The physical and spectroscopic properties of Ethyl 3-{[4-(8-sulfamoylquinolyl)methyl]-1*H*-1,2,3-triazol-1-yl}propanoate (**13a**) were consistent with literature data [20].

Ethyl 3-{[4-(*N*-methylsulfamoyl-8-quinolyl)methyl]-1*H*-1,2,3-triazol-1-yl}propanoate (**13b**):

Yield, 81 %; 110–111 °C. ^1^H NMR (DMSO-d_6_ and 300 MHz) δ, 1.15 (t, J = 7.2 Hz, 3H, and H-17); 2.79 (s, 3H, and H-9); 2.92 (t, J = 6.9 Hz, 2H, and H-13); 4.05 (q, J = 7.2 Hz, 2H, and H-16); 4.53 (t, J = 6.9 Hz, 2H, and H-14); 4.62 (s, 2H, and H-10); 7.68–7.79 (m, 2H, H-3, and H-6); 7.95 (s, 1H, and H-12); 8.31 (dd, J = 8.1 Hz, J = 1.2 Hz, 1H, and H-5); 8.39 (dd, J = 7.5 Hz, J = 1.2 Hz, 1H, and H-7); 8.54 (dd, J = 8.1 Hz, J = 1.5 Hz, 1H, and H-4); 9.08 (dd, J = 4.2 Hz, J = 1.5 Hz, 1H, and H-2). ^13^C NMR (DMSO-d_6_ and 75 MHz) δ: 14.5 (C-17), 34.4 (C-13), 35.3 (C-10), 45.6 (C-14), 46.1 (C-9), 60.8 (C-16), 122.9 (C-3), 124.1 (C-12), 126.2 (C-6), 129.1 (C-4a), 133.1 (C-7), 134.2 (C-5), 136.8 (C-8), 137.3 (C-4), 143.6 (C-11), 143.7 (C-8a), 151.9 (C-2), 170.7 (C-15). HRMS (ESI) *m*/*z*: C_18_H_22_N_5_O_4_S [M + H]^+^; calcd., 404.1392; found, 404.1387.

8-*N*-({1-[3β, 28-Diacetoxylup-20(29)-en-30-yl]-1*H*-1,2,3-triazol-4-yl}methyl)-quinolinesulfonamide (**14a**):

Yield, 75 %; m.p., 148–149 °C. ^1^H NMR (CDCl_3_ and 300 MHz) δ, 0.79–0.86 (m, 10H, H-5′, H-24′, H-23′, and H-25′); 0.97 (s, 3H, and H-27′); 1.03 (s, 3H, and H-26′); 1.05–1.32 (m, 8H, H-1′, H-6′, H-9′, H-11′, H-12′, H-15, H-16′, and H-22′); 1.40–1.50 (m, 5H, H-6′, H-7′, H-7′, H-11′, H-15′, and H-16′); 1.63–1.89 (m, 10H, H-1′, H-2′, H-2′, H-12′, H-13′, H-16′, H-18′, H-21′, H-22′, and H-22′); 2.05 (s, 3H, and H-34′); 2.06 (s, 3H, and H-32′); 2.27–2.28 (m, 1H, and H-19′); 3.76 (d, J = 10.8 Hz, 1H, and H-28′); 4.21–4.24 (m, 3H, H-10, and H-28′); 4.46–4.51 (m, 2H, H-3′, and H-29′); 4.74–4.88 (m, 2H, C-10, and H-30′); 5.00 (s, 1H, and H-29′); 6.91 (br, 1H, and H-9); 7.50 (s, 1H, and H-12); 7.59, (dd, t, J = 8.4 Hz, J = 4.2 Hz, 1H, and H-3); 7.69 (dd, J = 8.4 Hz, J = 7.5 1H, and H-6); 8.10 (dd, J = 8.4 Hz, J = 1.2 Hz, 1H, and H-5); 8.31 (dd, J = 8.4 Hz, J = 1.5 Hz, 1H, and H-4); 8.45 (dd, J = 7.5 Hz, J = 1.2 Hz, 1H, and H-7); 9.05 (dd, J = 4.2 Hz, J = 1.5 Hz, 1H, and H-2). ^13^C NMR (CDCl_3_ and 75 MHz) δ: 14.7 (C-27′), 16.0 (C-26′), 16.2 (C-24′), 16.5 (C-25′), 18.1 (C-6′), 20.9 (C-11′), 21.1 (C-32′), 21.4 (C-34′) 23.7 (C-2′), 26.8 (C-12′), 26.9 (C-15′), 28.0 (C-23′), 29.7 (C-16′), 31.2 (C-21′), 34.1 (C-7′), 34.3 (C-22′), 37.1 (C-10′), 37.4 (C-13′), 37.8 (C-4′), 38.4 (C-1′), 39.2 (C-9), 40.9 (C-8′), 42.7 (C-14′), 43.8 (C-19), 46.3 (C-17′), 50.0 (C-18′), 50.2 (C-9′), 54.4 (C-30′), 55.3 (C-5), 62.4 (C-28′), 80.9 (C-3′), 112.0 (C-29′), 122.5 (C-3), 122.7 (C-12), 125.7 (C-6), 128.8 (C-4a), 131.2 (C-7), 133.5 (C-5), 135.6 (C-4), 137.2 (C-8), 142.9 (C-8a), 144.5 (C-11), 148.8 (C-20′), 151.4 (C-2), 171.1 (C-33′), 171.6 (C-31′). HRMS (ESI) *m*/*z*: C_46_H_64_N_5_O_6_S [M + H]^+^; calcd., 814.4577; found, 814.4591.

The physical and spectroscopic properties of 8-*N*-methyl-*N*-({1-[3β, 28-diacetoxylup-20(29)-en-30-yl]-1*H*-1,2,3-triazol-4-yl}methyl)-quinolinesulfonamide (**14b**) were consistent with literature data [20].

### 3.3. In Vitro Studies

Cell cultures were performed in 96-well plates (Nunc Thermo Fisher Scientific, Waltham, MA, USA). Cells were seeded at 5 × 104/well and incubated for 24 h (at 37 °C, 5% CO_2_, and constant humidity). The medium was replaced with fresh with the addition of test compounds at a concentration of 0.1–100 µg/mL of DMSO and incubated again for another 72 h. After this time, the WST-1 test (Roche Molecular Biochemicals, Mannheim, Germany) was performed to assess the metabolic activity of the cells. Absorbance was measured at λ = 450 nm using a UVM340 microplate reader (Biogenet, Józefów, Poland). Results are expressed as the mean value of at least three independent experiments performed in triplicate.

## 4. Conclusions

In this work, a series of 8-quinolinesulfonamide and 1,4-disubstituted triazole derivatives were designed and synthesised. The first stage of in silico work used machine learning, molecular docking, and molecular dynamics. The results obtained using these computational techniques allowed for the preliminary identification of six derivatives that may have antiproliferative activity: **9a**, **9b**, **10a**, **10b**, **11a**, and **14b**. The ADMET profile calculated for these compounds was also favourable. In the second part of this article, activity tests were performed against four cancer cell lines and normal cells. The results of in silico and in vitro experiments allowed for the selection of derivative **9b** (8-*N*-methyl-*N*-[1-(7-chloroquinolin-4-yl)-1*H*-1,2,3-triazol-4-yl]methyl quinolinesulfonamide) as a leading structure in further research on the anticancer activity of hybrids of quinolinesulfonamides and triazoles.

## Data Availability

All data are available on request from the corresponding author.

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
