# Peer review of "Synthesis, Docking, and Machine Learning Studies of Some Novel Quinolinesulfonamides–Triazole Hybrids with Anticancer Activity"

_molecules, 2024, doi:10.3390/molecules29133158_

Round 1

Reviewer 1 Report

Comments and Suggestions for Authors

Dear Editor,

The manuscript by Krzysztof Marciniec et al. describes the synthesis, docking and machine learning studies of 22 hybrids of 8-quinolinesulfonamide and 1,4-disubstituted triazole with antiproliferative activity. The new molecules are well characterized and some of them seem promising candidates as anticancer drugs. However, the manuscript is very difficult to read because sometime there is not a clear conclusion for each experiment for the design and in silico prediction of ROCK1 inhibitors. Furthermore, some points should be improved or clarified before publishing this manuscript:

1.       Figure 2 seems incomplete. C should be identified in the figure.

2.       The rational design of the library of quinolinesulfamoyl compounds it is not clear, what was the criteria for the selection of the derivatives?

3.       It is stated that “From an in-house library of quinolinesulfamoyl compounds, 22 derivatives were selected for synthesis (Figure 3) using ML aproaches.” The list of compunds in that library should be provided.

4.       AutoDock Vina program indicates that compounds 9b, 10a and 10b seem promising candidates, however only compunds 9b was studied using Molecular docking and Molecular dynamics calculations. More details should be provided on that decision.

5.       Conclusions point out “The results obtained using these computational techniques allowed for the preliminary identification of six derivatives that may have antiproliferative activity.” Those compounds should be indicated.

Author Response

POINT-BY-POINT RESPONSES TO REVIEWER 1.

Dear Editor,

The manuscript by Krzysztof Marciniec et al. describes the synthesis, docking and machine learning studies of 22 hybrids of 8-quinolinesulfonamide and 1,4-disubstituted triazole with antiproliferative activity. The new molecules are well characterized and some of them seem promising candidates as anticancer drugs. However, the manuscript is very difficult to read because sometime there is not a clear conclusion for each experiment for the design and in silico prediction of ROCK1 inhibitors. Furthermore, some points should be improved or clarified before publishing this manuscript:

RESPONSE: We would like to thank the Reviewer for the assessment of our manuscript and the opinion that our paper.

  1. Figure 2 seems incomplete. C should be identified in the figure.

RESPONSE: Good point. We corrected and we have completed part C of Figure 2.

  1. The rational design of the library of quinolinesulfamoyl compounds it is not clear, what was the criteria for the selection of the derivatives?

RESPONSE: As requested by the reviewer, the following paragraph explaining the criteria for selecting sulfamoylquinoline derivatives as the primary pharmacological system in this work has been added at line 56:

 “It should be mentioned that a large group of ROCK inhibitors contains an azine system in its structure, among others. pyridine, isoquinoline, quinoline, or quinazoline. And some of the sulfamoyl derivatives of azines, which are ROCK inhibitors, have been approved in Japan and/or China (fasudil and ripasudil).”

  1. It is stated that “From an in-house library of quinolinesulfamoyl compounds, 22 derivatives were selected for synthesis (Figure 3) using ML aproaches.” The list of compunds in that library should be provided.

RESPONSE: Compound structures have been added to the supplement (Table 1S)

  1. AutoDock Vina program indicates that compounds 9b, 10a and 10b seem promising candidates, however only compunds 9b was studied using Molecular docking and Molecular dynamics calculations. More details should be provided on tat decision.

RESPONSE: Thank you for this suggestion. We have added detailed analysis also for compounds 10a and 10b

  1. Conclusions point out “The results obtained using these computational techniques allowed for the preliminary identification of six derivatives that may have antiproliferative activity.” Those compounds should be indicated.

RESPONSE: Done.

Reviewer 2 Report

Comments and Suggestions for Authors

The article is devoted to synthesis of new different derivatives, which were evaluated for anticancer activity. The article is interesting because it presents results useful to scientists working in the fields of medicine and natural chemistry. The article is well presented, and the illustrations complement the text. The authors have provided comprehensive information on the methods of synthesis and biological evaluation of the compounds. The conclusions are consistent with the evidence and arguments presented. However, article can be accepted to publish after minor revisions: 

1. Will be better if compound 9b will be mentioned in abstract as name instead number 

2. Chemistry part: method ii - the amount of propargyl alcohol must be specified. In this reaction, a disubstitution product cannot be obtained? Please give some explanation. Procedures A or B - the ratio of solvents must be given. 

Other minor notes:

Line 52: a definition needs to be given for RhoA/ROCK 

Line 63 extra comma after «The ROCK family». Should be deleted

All references in the text should be given as a range, not as a list (i.e. [4],[5],[6] should be written as 4-6)

In silico in vitro etc. should be italic 

Comments on the Quality of English Language

Minor editing of English language required

Author Response

POINT-BY-POINT RESPONSES TO REVIEWER 2

The article is devoted to synthesis of new different derivatives, which were evaluated for anticancer activity. The article is interesting because it presents results useful to scientists working in the fields of medicine and natural chemistry. The article is well presented, and the illustrations complement the text. The authors have provided comprehensive information on the methods of synthesis and biological evaluation of the compounds. The conclusions are consistent with the evidence and arguments presented. However, article can be accepted to publish after minor revisions:

RESPONSE: Than You

  1. Will be better if compound 9b will be mentioned in abstract as name instead number

RESPONSE: We changed the compound number in the abstract to its name.

  1. Chemistry part: method ii - the amount of propargyl alcohol must be specified. In this reaction, a disubstitution product cannot be obtained? Please give some explanation.

RESPONSE: Appropriate synthetic information has been added to the text.

Procedures A or B - the ratio of solvents must be given. 

RESPONSE: Procedures A and B have been improved. The required information has been added

Other minor notes:

Line 52: a definition needs to be given for RhoA/ROCK 

RESPONSE: As requested by the reviewer, the ROCK definition has been moved from line 57 to line 52

Line 63 extra comma after «The ROCK family». Should be deleted

RESPONSE: Done

All references in the text should be given as a range, not as a list (i.e. [4],[5],[6] should be written as 4-6)

RESPONSE: Thank you for this suggestion. We have changed the citation format.

In silico in vitro etc. should be italic

RESPONSE: Done

Reviewer 3 Report

Comments and Suggestions for Authors

Thank you for submitting your manuscript titled "Synthesis, Docking, and Machine Learning Studies of Some Novel Quinolinesulfonamides-triazole Hybrids with Anti-cancer Activity" to Molecules. The study is well-constructed and presents significant findings on the design, synthesis, and biological evaluation of novel quinolinesulfonamide-triazole hybrids. These compounds were assessed for their anti-cancer activity using a combination of molecular docking, machine learning, and in vitro assays. The most promising compound, 9b, exhibited high cytotoxicity against selected cancer cell lines without affecting normal cells. While the manuscript is promising, several minor revisions are needed to improve clarity and coherence:

1. The abstract is too condensed and lacks clarity. This section contains numerous abbreviations and technical jargon that are not clearly defined, like “ADMET” in Line 14 and “Compound 9b” in Line 16. Compound 9b is one of your samples, while it cannot be understood in abstract part for audience.

2. The introduction section needs better organization and clarity. Simplify the language and provide clear definitions for abbreviations the first time they are used, for example, the definition of ROCK should be provided in Line 52 rather than Line 57. In addition, expand on the significance of quinolinesulfonamides and triazoles in cancer therapy with recent references.

3. Provide a clear explanation of the machine learning methods and the specific software or algorithms used. Avoid assuming the reader is familiar with all terms and methodologies. For example, in Line 127, the authors mentioned about the choice of cut-off value but they didn’t provide their solutions about how to choose a good cut-off.

4. In Figure 2, some examples of Part C can be provided in this figure.

By addressing these minor issues, the manuscript will be clearer and more impactful. I look forward to seeing the revised version.

Author Response

POINT-BY-POINT RESPONSES TO REVIEWER 3.

Thank you for submitting your manuscript titled "Synthesis, Docking, and Machine Learning Studies of Some Novel Quinolinesulfonamides-triazole Hybrids with Anti-cancer Activity" to Molecules. The study is well-constructed and presents significant findings on the design, synthesis, and biological evaluation of novel quinolinesulfonamide-triazole hybrids. These compounds were assessed for their anti-cancer activity using a combination of molecular docking, machine learning, and in vitro assays. The most promising compound, 9b, exhibited high cytotoxicity against selected cancer cell lines without affecting normal cells. While the manuscript is promising, several minor revisions are needed to improve clarity and coherence:

RESPONSE: We would like to thank the Reviewer for the assessment of our manuscript and the opinion that our paper

  1. The abstract is too condensed and lacks clarity. This section contains numerous abbreviations and technical jargon that are not clearly defined, like “ADMET” in Line 14 and “Compound 9b” in Line 16. Compound 9b is one of your samples, while it cannot be understood in abstract part for audience.

RESPONSE: The abstract has been corrected. An unclear abbreviation (ADMET) was removed and a shortened version of the name of compound 9b specifying its structure was added. The high degree of condensation of the abstract results from the fact that, according to the editorial guidelines, a lot of information should be included in 200 words.

  1. The introduction section needs better organization and clarity. Simplify the language and provide clear definitions for abbreviations the first time they are used, for example, the definition of ROCK should be provided in Line 52 rather than Line 57. In addition, expand on the significance of quinolinesulfonamides and triazoles in cancer therapy with recent references.

RESPONSE: Thank you for this suggestion. We have made the suggested changes to the text. We have supplemented the literature on the anticancer activity of quinolinesulfonamides. We have also added information on the anticancer activity of triazoles.

  1. Provide a clear explanation of the machine learning methods and the specific software or algorithms used. Avoid assuming the reader is familiar with all terms and methodologies. For example, in Line 127, the authors mentioned about the choice of cut-off value but they didn’t provide their solutions about how to choose a good cut-off.

RESPONSE:

Thank you for this suggestion. The description of the Machine Learning method has been simplified.

  1. In Figure 2, some examples of Part C can be provided in this figure.

RESPONSE: ROCK inhibitors are very diverse and contain a significant number of substituents and functional groups. Therefore, we have supplemented Figure 2 with general information regarding the type of substituents present in part C.

By addressing these minor issues, the manuscript will be clearer and more impactful. I look forward to seeing the revised version.

RESPONSE: Than you.

Round 2

Reviewer 1 Report

Comments and Suggestions for Authors

Dear Editor,

The manuscript by Krzysztof Marciniec et al. describes the synthesis, docking and machine learning studies of 22 hybrids of 8-quinolinesulfonamide and 1,4-disubstituted triazole with antiproliferative activity. After revisions, the manuscript can be read more fluently and would be more attractive for researchers. All the issues have been addressed point by point and clarified, so in my opinion this work could be published as it is.

Reviewer 3 Report

Comments and Suggestions for Authors

After revision, the manuscript is accepted for publication.